# The Behavior of Melts with Vanishing Viscosity in the Cone-and-Plate Rheometer

**Lihui Lang [1], Sergei Alexandrov [1,2], Elena Lyamina [3,4,*] and Van Manh Dinh [5]**

[1] School of Mechanical Engineering and Automation, Beihang University, Beijing 100191, China; lang@buaa.edu.cn (L.L.); sergei_alexandrov@spartak.ru (S.A.)

[2] Ishlinsky Institute for Problems in Mechanics RAS, Moscow 119526, Russia

[3] Division of Computational Mathematics and Engineering, Institute for Computational Science, Ton Duc Thang University, Ho Chi Minh City 700000, Vietnam

[4] Faculty of Civil Engineering, Ton Duc Thang University, Ho Chi Minh City 700000, Vietnam

[5] Institute of Mechanics, VAST, 18 Hoang Quoc Viet, Hanoi 700000, Vietnam; dinhvanmanh@gmail.com

[*] Correspondence: lyaminaea@tdtu.edu.vn; Tel.: +842837755024

**Abstract:** A semi-analytic solution for material flow in the cone-and-plate rheometer is presented. It is assumed that the viscosity is solely a function of the second invariant of the strain rate tensor. A distinguishing feature of the constitutive equations used is that the viscosity is vanishing as the shear strain rate approaches infinity. This feature of the constitutive equations affects the qualitative behavior of the solution. Asymptotic analysis is carried out near the surface of the cone to reveal these features. It is shown that the regime of sliding must occur and the shear strain rate approaches infinity under certain conditions. It is also shown that the asymptotic behavior of the viscosity as the shear strain rate approaches infinity controls these qualitative features of the theoretical solution. Some of these features are feasible for experimental verification. An interpretation of the theoretical solution found is proposed.

**Keywords:** vanishing viscosity; rough wall; slip; singularity; continuum mechanics

---

## 1. Introduction

Viscosity, defined as the ratio of the shear stress to shear strain rate in a steady simple shear flow, usually depends on the shear strain rate. The constitutive equations of many materials including polymers assume that the viscosity is vanishing as the shear strain rate approaches infinity [1–8]. This feature of the constitutive equations may have a qualitative effect on solution behavior near rough walls, in particular in the description of material flow in rheometers.

The cone-and-plate rheometer is widely used to determine viscosity [3,4,9–14]. Theoretical descriptions of material flow in this rheometer have been provided in [15,16]. In the case of small cone angles, the flow in the cone-and-plate rheometer is considered to be ideal in the sense that it is practically homogeneous [11]. For this reason, a new description of the flow in this rheometer is proposed in the present paper to show the effect of vanishing viscosity on the interpretation of experimental data.

It is shown that the qualitative behavior of the solution for the flow in the cone-and-plate rheometer depends on the asymptotic behavior of the shear stress as the shear strain rate approaches infinity. In particular, there are such dependencies of the shear stress on the shear strain rate that no solution at sticking at the cone exists. It is therefore possible to describe the interface behavior using special constitutive equations in a narrow region near the wall. It is notable that this feature of the solution is very different from numerous discussions on the slip boundary condition [6,17–29].

In particular, there is a vast amount of literature on experimental studies related to the wall slip phenomenon [17–21,23–25]. Many theoretical works are devoted to the apparent slip mechanism, for example in [27,29]. Another theoretical model for the boundary slip has been proposed in [22]. It is seen from these papers and from review papers [26,28] that the possibility to describe the occurrence of slip using appropriate constitutive equations in the vicinity of the wall has not been considered. In the present paper, it is shown that under certain conditions the transition between the regimes of sticking and sliding is described by the constitutive equations rather than by the friction law.

The solution neglects normal stress effects. Therefore, its practical use is restricted to materials for which this assumption is adequate.

The physical components of tensors in a spherical coordinate system are used throughout the paper.

## 2. Basic Equations and Assumptions for Cone-and-Plate Rheometer

A schematic diagram of cone-and-plate rheometers is shown in Figure 1. The circular disk is motionless. The cone rotates with angular velocity $\omega$. The process has an axis of symmetry.

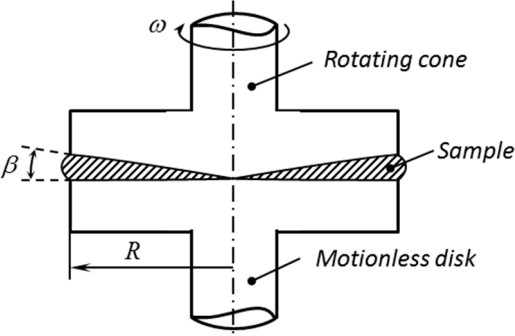

**Figure 1.** Cone-and-plate rheometer – notation.

End and inertia effects are neglected. Then, the material flow between the disk and cone can be approximated by the material flow between two coordinate surfaces, $\theta = \pi/2$ and $\theta = \pi/2 - \beta = \theta_0$, of a spherical coordinate system $(r, \theta, \varphi)$ whose axis $\theta = 0$ coincides with the axis of symmetry of the process (Figure 2). It is evident that the solution is independent of $\varphi$. Those are typical assumptions used for describing the flow in the cone-and-plate rheometer [16].

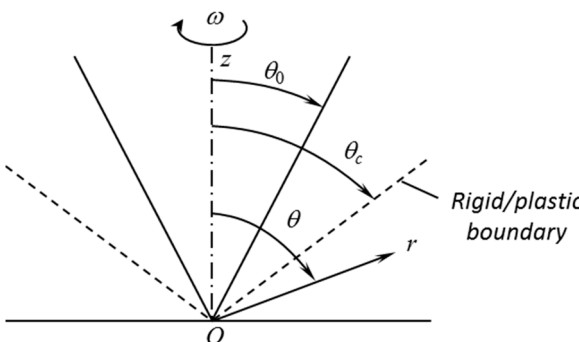

**Figure 2.** Illustration of the boundary value problem.

Let $\sigma_{rr}, \sigma_{\theta\theta}, \sigma_{\varphi\varphi}, \sigma_{r\theta}, \sigma_{r\varphi}$, and $\sigma_{\theta\varphi}$ be the stress components referred to the spherical coordinate system. It is assumed that

$$\sigma_{rr} = \sigma_{\theta\theta} = \sigma_{\varphi\varphi} = \sigma, \ \sigma_{r\theta} = \sigma_{r\varphi} = 0. \tag{1}$$

This assumption should be verified a posteriori. It is evident that $\sigma$ is the hydrostatic stress. Moreover, it is assumed that

$$u_r = u_\theta = 0, \ u_\varphi = u \tag{2}$$

where $u_r$, $u_\theta$ and $u_\varphi$ are the velocity components referred to the spherical coordinate system. This assumption should also be verified a posteriori. The direction of rotation of the cone (Figures 1 and 2) demands that

$$u < 0 \text{ and } \sigma_{\theta\varphi} > 0. \tag{3}$$

Under the assumptions above, the equations of equilibrium referred to the spherical coordinate system read

$$\frac{\partial \sigma}{\partial r} = 0, \ \frac{\partial \sigma}{\partial \theta} = 0, \ \frac{\partial \sigma_{\theta\varphi}}{\partial \theta} + 2\sigma_{\theta\varphi} \cot \theta = 0. \tag{4}$$

The first two equations and the independence of the solution of $\varphi$ demand that

$$\sigma = \sigma_0 \tag{5}$$

where $\sigma_0$ is constant.

Using (2) one can represent the components of the strain rate tensor in the spherical coordinate system as

$$\xi_{rr} = 0, \ \xi_{\theta\theta} = 0, \ \xi_{\varphi\varphi} = 0, \ \xi_{r\theta} = 0, \ \xi_{r\varphi} = \frac{1}{2}\left(\frac{\partial u}{\partial r} - \frac{u}{r}\right), \ \xi_{\theta\varphi} = \frac{1}{2r\sin\theta}\left(\sin\theta\frac{\partial u}{\partial \theta} - u\cos\theta\right). \tag{6}$$

The constitutive equations are the yield criterion

$$(\sigma_{rr} - \sigma_{\theta\theta})^2 + (\sigma_{\theta\theta} - \sigma_{\varphi\varphi})^2 + (\sigma_{\varphi\varphi} - \sigma_{rr})^2 + 6(\sigma_{r\theta}^2 + \sigma_{\theta\varphi}^2 + \sigma_{\varphi r}^2) = 6k_0^2\Phi^2\left(\frac{\xi_{eq}}{\xi_0}\right) \tag{7}$$

and its associated flow rule

$$\xi_{rr} = \lambda(2\sigma_{rr} - \sigma_{\theta\theta} - \sigma_{\varphi\varphi}), \ \xi_{\theta\theta} = \lambda(2\sigma_{\theta\theta} - \sigma_{rr} - \sigma_{\varphi\varphi}), \ \xi_{\varphi\varphi} = \lambda(2\sigma_{rr} - \sigma_{\theta\theta} - \sigma_{\varphi\varphi}),$$
$$\xi_{r\theta} = 6\lambda\sigma_{r\theta}, \ \xi_{r\varphi} = 6\lambda\sigma_{r\varphi}, \ \xi_{\theta\varphi} = 6\lambda\sigma_{\theta\varphi}. \tag{8}$$

where $\lambda$ is a non-negative multiplier, $\xi_{eq}$ is the equivalent strain rate, and $\xi_0$ is a reference strain rate. Also, $\Phi(\xi_{eq}/\xi_0)$ is an arbitrary monotonically increasing function of its argument and $k_0$ is the shear yield stress at $\xi_{eq} = 0$. The former means that the solution below is restricted to materials whose response increases as the magnitude of the equivalent strain rate increases. The latter means that $\Phi(0) = 1$. Both $\xi_0$ and $k_0$ are material constants. It is convenient to define the equivalent strain rate as

$$\xi_{eq} = \frac{1}{\sqrt{2}}\sqrt{\xi_{rr}^2 + \xi_{\theta\theta}^2 + \xi_{\varphi\varphi}^2 + 2\xi_{r\theta}^2 + 2\xi_{\theta\varphi}^2 + 2\xi_{\varphi r}^2}. \tag{9}$$

Substituting (1) into (8) leads to

$$\xi_{rr} = 0, \ \xi_{\theta\theta} = 0, \ \xi_{\varphi\varphi} = 0, \ \xi_{r\theta} = 0, \text{ and } \xi_{r\varphi} = 0. \tag{10}$$

The first four equations are compatible with (6). The fifth equation and Equation (6) for $\xi_{r\varphi}$ combine to give

$$u = -\omega r w(\theta) \tag{11}$$

where $w(\theta)$ is an arbitrary function of $\theta$. It follows from (3) and (11) that $w(\theta) > 0$. The only restriction on the solution imposed by the last equation in (8) is that the sign of $\xi_{\theta\varphi}$ coincides with the sign of $\sigma_{\theta\varphi}$. Then, it follows from (3) that

$$\xi_{\theta\varphi} > 0. \tag{12}$$

Thus the associated flow rule is satisfied if (12) is satisfied. Substituting (10) into (9), and taking into account (12), one can get

$$\xi_{eq} = \xi_{\theta\varphi}. \tag{13}$$

Turning to the yield criterion (7), this criterion, (1) and (3) combine to give

$$\sigma_{\theta\varphi} = k_0 \Phi\left(\frac{\xi_{eq}}{\xi_0}\right). \tag{14}$$

Using (13) this equation can be rewritten as

$$\sigma_{\theta\varphi} = k_0 \Phi\left(\frac{\xi_{\theta\varphi}}{\xi_0}\right) \tag{15}$$

giving the relationship between the shear stress and shear strain rate in the spherical coordinate system. The viscosity is usually defined as (in our nomenclature)

$$\eta = \frac{\sigma_{\theta\varphi}}{\xi_{\theta\varphi}}. \tag{16}$$

In the case of materials with vanishing viscosity, $\eta \to 0$ as $\xi_{\theta\varphi} \to \infty$. Then, it follows from (15) and (16) that

$$\frac{\Phi\left(\xi_{\theta\varphi}/\xi_0\right)}{\xi_{\theta\varphi}/\xi_0} \to 0 \tag{17}$$

as $\xi_{\theta\varphi} \to \infty$. Henceforward, attention is focused on the class of functions that are represented as

$$\Phi\left(\frac{\xi_{\theta\varphi}}{\xi_0}\right) = \frac{k_s}{k_0} + A\left(\frac{\xi_{\theta\varphi}}{\xi_0}\right)^\alpha + o\left[\left(\frac{\xi_{\theta\varphi}}{\xi_0}\right)^\alpha\right] \tag{18}$$

as $\xi_{\theta\varphi} \to \infty$. Here $k_s$, $A$ and $\alpha$ are constant. It is evident that this class is large enough for any function $\Phi\left(\xi_{eq}\right)$ used in applications. It follows from the representation (18) that (17) is satisfied if

$$\alpha < 1. \tag{19}$$

Since $\Phi$ is a monotonically increasing function of its argument, $A > 0$ if $\alpha > 0$ and $A < 0$ if $\alpha < 0$. The function $\Phi$ is independent of the shear strain rate if $\alpha = 0$. The response of such materials is independent of the speed of loading. An example is the rigid perfectly plastic material [30]. This special case is not considered in the present paper. If $\alpha > 0$, then the term $k_s/k_0$ in (18) is negligible as $\xi_{\theta\varphi} \to \infty$. If $\alpha < 0$, then $k_s$ is the maximum possible shear stress that may appear in the material. Equation (18) can be solved for $\xi_{\theta\varphi}$ to give

$$\frac{\xi_{\theta\varphi}}{\xi_0} = \begin{cases} A^{-1/\alpha}\left(\frac{\sigma_{\theta\varphi}}{k_0}\right)^{1/\alpha} + o\left[\left(\frac{\sigma_{\theta\varphi}}{k_0}\right)^{1/\alpha}\right] & \text{as } \sigma_{\theta\varphi} \to \infty \text{ if } \alpha > 0 \\ \left[-\frac{1}{A}\left(\frac{k_s}{k_0} - \frac{\sigma_{\theta\varphi}}{k_0}\right)\right]^{1/\alpha} + o\left[\left(\frac{k_s}{k_0} - \frac{\sigma_{\theta\varphi}}{k_0}\right)^{1/\alpha}\right] & \text{as } \sigma_{\theta\varphi} \to k_s \text{ if } \alpha < 0 \end{cases} \tag{20}$$

It remains to solve the third equation in (4) together with (11) and (15). The boundary conditions are initially imposed on the velocity $u$ and require the regime of sticking at both contact surfaces, $\theta = \pi/2$ and $\theta = \theta_0$ (Figure 2). Using (11), these conditions can be represented in terms of the function $w(\theta)$ as

$$w = 0 \tag{21}$$

for $\theta = \pi/2$ and

$$w = \sin\theta_0 \tag{22}$$

for $\theta = \theta_0$. However, as it will be seen later, a difficulty is that no solution satisfying the boundary conditions (21) and (22) may exist. In this case, it is necessary to assume that the regime of sliding occurs at one of the contact surfaces. Therefore, in contrast to conventional approaches, the transition between the regimes of sticking and sliding is described by the constitutive equations rather than by the friction law. Moreover, a rigid region may appear. In this case, one of the boundary conditions, (21) or (22), should be satisfied at the rigid plastic boundary rather than at the contact surface.

## 3. Solution

The third equation in (4) can be immediately integrated to give

$$\sigma_{\theta\varphi} = \frac{C_1 k_0}{\sin^2 \theta} \tag{23}$$

where $C_1$ is constant. It follows from (3) that $C_1 > 0$. Equations (15) and (23) combine to yield

$$\frac{\xi_{\theta\varphi}}{\xi_0} = \Lambda\left(\frac{C_1}{\sin^2 \theta}\right) \tag{24}$$

where $\Lambda$ is the function inverse to $\Phi$. Using (6) and (11) the shear strain rate is expressed as

$$\xi_{\theta\varphi} = -\frac{\omega}{2}\left(\frac{dw}{d\theta} - w \cot \theta\right). \tag{25}$$

Eliminating $\xi_{\theta\varphi}$ in (24) by means of (25) results in

$$\frac{dw}{d\theta} - w \cot \theta = -\mu\Lambda\left(\frac{C_1}{\sin^2 \theta}\right) \tag{26}$$

where $\mu = 2\xi_0/\omega$. The solution of this equation satisfying the boundary condition (21) is

$$w = \mu \sin \theta \int_{\theta}^{\pi/2} \frac{1}{\sin \chi} \Lambda\left(\frac{C_1}{\sin^2 \chi}\right) d\chi. \tag{27}$$

The equation for $C_1$ follows from (22) and (27) in the form

$$1 = \mu \int_{\theta_0}^{\pi/2} \frac{1}{\sin \theta} \Lambda\left(\frac{C_1}{\sin^2 \theta}\right) d\theta. \tag{28}$$

Equations (27) and (28) are valid if there is no rigid region. It is seen from (23) that if a rigid region exists then it is adjacent to the contact surface $\theta = \pi/2$. It is assumed that the rigid/plastic boundary is determined by the equation $\theta = \theta_c$ (Figure 2). In this case, the boundary condition (21) should be replaced with

$$w = 0 \tag{29}$$

for $\theta = \theta_c$. Since $\sigma_{\theta\varphi} = k_0$ at the rigid/plastic boundary, it follows from (23) that

$$C_1 = \sin^2 \theta_c. \tag{30}$$

Then, Equations (27) and (28) become

$$w = \mu \sin \theta \int_{\theta}^{\theta_c} \frac{1}{\sin \chi} \Lambda\left(\frac{\sin^2 \theta_c}{\sin^2 \chi}\right) d\chi \tag{31}$$

and

$$1 = \mu \int_{\gamma}^{\theta_c} \frac{1}{\sin \theta} \Lambda\left(\frac{\sin^2 \theta_c}{\sin^2 \theta}\right) d\theta. \tag{32}$$

Equation (32) serves for determining $\theta_c$.

It is seen from the definition for $\mu$ that $\mu \to 0$ as $\omega \to \infty$. Therefore, no solution to Equation (28) or (32) may exist.

## 4. Asymptotic Analysis

A necessary condition to satisfy Equation (28) or (32) as $\mu \to 0$ is $\Lambda \to \infty$ at some point of the interval $\theta_0 \le \theta \le \pi/2$ or $\theta_0 \le \theta \le \theta_c$, respectively. By assumption, $\Phi$ is a monotonically increasing function of its argument. It is seen from (23) that the argument is a monotonically decreasing function of $\theta$. Therefore, $\Lambda$ may approach infinity only in the vicinity of $\theta = \theta_0$.

Equation (28) can be rewritten as

$$1 = \mu I_1 + \mu I_2 \tag{33}$$

where

$$I_1 = \int_{\theta_0}^{(1+\delta)\theta_0} \frac{1}{\sin \theta} \Lambda\left(\frac{C_1}{\sin^2 \theta}\right) d\theta, \quad I_2 = \int_{(1+\delta)\theta_0}^{\pi/2} \frac{1}{\sin \theta} \Lambda\left(\frac{C_1}{\sin^2 \theta}\right) d\theta \tag{34}$$

and $0 < \delta << 1$. If $\alpha > 0$ then it follows from (20) and (23) that

$$\Lambda\left(\frac{C_1}{\sin^2 \theta}\right) = A^{-1/\alpha}\left(\frac{C_1}{\sin^2 \theta}\right)^{1/\alpha} \tag{35}$$

in the vicinity of the surface $\theta = \theta_0$. It is then seen from (34) that $I_1 \to \infty$ as $C_1 \to \infty$. Therefore, Equation (34) has a solution at any small value of $\mu$ and the regime of sticking at the surface $\theta = \theta_0$ is always possible.

If $\alpha < 0$, then it follows from (20) and (23) that

$$\Lambda\left(\frac{C_1}{\sin^2 \theta}\right) = \left[-\frac{1}{A}\left(\frac{k_s}{k_0} - \frac{C_1}{\sin^2 \theta}\right)\right]^{1/\alpha} \tag{36}$$

in the vicinity of the surface $\theta = \theta_0$. It is seen from this equation that the maximum possible value of $C_1$ is

$$C_m = \frac{k_s \sin^2 \theta_0}{k_0}. \tag{37}$$

Therefore, the integral $I_2$ is bounded at any value of $C_1$. Using (36) and (37) the integral $I_1$ at $C_1 = C_m$ can be written as

$$I_1 = \left(-\frac{1}{A}\frac{k_s}{k_0}\right)^{1/\alpha} \int_{\theta_0}^{(1+\delta)\theta_0} \frac{1}{\sin \theta}\left[\left(1 - \frac{\sin^2 \theta_0}{\sin^2 \theta}\right)\right]^{1/\alpha} d\theta. \tag{38}$$

The integrand is represented as

$$\frac{1}{\sin\theta}\left[\left(1-\frac{\sin^2\theta_0}{\sin^2\theta}\right)\right]^{1/\alpha} = B(\theta-\theta_0)^{1/\alpha} + o\left[(\theta-\theta_0)^{1/\alpha}\right] \tag{39}$$

as $\theta \to \theta_0$. Here $B$ is a function of $\theta_0$. It is evident from (38) and (39) that the integral $I_1$ is divergent if

$$\frac{1}{\alpha}+1 \le 0 \tag{40}$$

and convergent if

$$\frac{1}{\alpha}+1 > 0 \tag{41}$$

Therefore, the regime of sticking at the surface $\theta = \theta_0$ is always possible if inequality (40) is satisfied. On the other hand, the regime of sliding occurs at $C_1 = C_m$ if inequality (41) is satisfied. It will be seen later that this qualitative difference in solution behavior predicts the qualitative difference in the dependence between two measurable quantities, $\omega$ and the torque.

It is seen from (25) and (26) that $\xi_{\theta\varphi}$ is proportional to $\Lambda$. Then, it follows from (39) that

$$\xi_{\theta\varphi} = O\left[(\theta-\theta_0)^{1/\alpha}\right] \tag{42}$$

as $\theta \to \theta_0$ if $C_1 = C_m$. It is seen from this equation that the velocity gradient approaches infinity near the friction surface if $\alpha < 0$. On the other hand, the shear stress is bounded everywhere, as follows from (18). Then, it is evident from (42) that the plastic work rate is given by

$$E = \xi_{\theta\varphi}\sigma_{\theta\varphi} = O\left[(\theta-\theta_0)^{1/\alpha}\right] \tag{43}$$

as $\theta \to \theta_0$. Then, the inequality (41) and (43) ensure that the triple integral $\iiint_V E dV$ converges. Here, $V$ is the volume of the sample.

The case corresponding to Equation (32) can be treated in a similar manner.

## 5. Interpretation of the Solution and Discussion

Using the geometry of Figure 1, the torque is found as

$$T = \frac{2\pi\tau R^3}{3\cos\beta}. \tag{44}$$

It has been taken into account here that $\sigma_{\theta\varphi}$ is independent of $r$, as follows from (23). Moreover, $\tau$ is the value of $\sigma_{\theta\varphi}$ at $\theta = \theta_0$. If the regime of sticking occurs then the value of $\tau$ depends on $\omega$ (or $\mu$). In particular, using (23), Equation (44) can be rewritten as

$$t = \frac{3T}{2\pi R^3 k_0} = \frac{C_1}{\cos^3\beta} \tag{45}$$

where $t$ is the dimensionless torque. It has been taken into account here that $\theta_0 = \pi/2 - \beta$, as it is seen from Figures 1 and 2. The value of $C_1$ is determined from (28) if there is no rigid region. If there is a rigid region then (3) should be used to transform (45) to

$$t = \frac{\sin^2\theta_c}{\cos^3\beta} \tag{46}$$

where $\theta_c$ is determined from (32).

If $\alpha$ satisfies the inequality (41) and $C_1$ attains the value of $C_m$, then Equations (37) and (45) combine to give

$$t_m = \frac{k_s}{k_0 \cos \beta}. \tag{47}$$

It is evident that the value of $t_m$ is independent of $\mu$ and is described by material properties and geometry.

If $\alpha$ satisfies the inequality (40) then $C_1 \to \infty$ as $\omega \to \infty$. Therefore, it is seen from Equation (45) that $t \to \infty$ as $\omega \to \infty$.

The dependence of $t$ on $\omega$ that follows from the discussion above is shown schematically in Figure 3 where curve 1 corresponds to a value of $\alpha$ satisfying the inequality (41) and curve 2 to a value of $\alpha$ satisfying the inequality (40). The qualitative behavior of these curves is independent of the function $\Phi\left(\xi_{eq}/\xi_0\right)$ over any finite interval $0 \leq \xi_{eq}/\xi_0 \leq \xi_m/\xi_0 < \infty$. In general, for some materials, experimental observations confirm the general features of the dependence between the torque and $\mu$ illustrated in Figure 3 by curve 1 (see, for example, Figure 8 in [26]). In particular, the torque is practically independent of $\omega$ in the range $\omega \geq \omega_m$ where $\omega_m = 2\xi_0/\mu_m$.

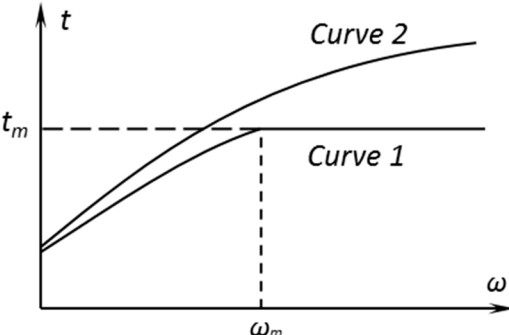

**Figure 3.** Schematic representation of the dependence of $t$ on $\omega$ for satisfying inequalities (41) and (40). Curve 1 corresponds to (41) and curve 2 to (40).

The value of $\mu_m$ is determined from (28) at $C_1 = C_m$. Then, using (37) one gets the following equation for $\mu_m$:

$$1 = \mu_m \int_{\theta_0}^{\pi/2} \frac{1}{\sin \theta} \Lambda\left(\frac{k_s}{k_0} \frac{\sin^2 \theta_0}{\sin^2 \theta}\right) d\theta.$$

A conventional interpretation of the dependence between the torque and $\omega$ illustrated in Figure 3 by curve 1 is that there is a transition region between weak and strong slip regions. The main result of the present paper provides another interpretation that the finite value of $k_s$ exists and that the value of $\alpha$ satisfies (41). In this case, experimental results similar to that illustrated in Figure 8 in [26] allow for the value of $k_s/k_0$ to be determined using Equation (47). All other parameters of the material model can be determined using any available method, for example [31].

In some cases, rough surfaces are intentionally used for suppressing slip in rheometers [32]. This method has no effect of material flow in the vicinity of interfaces if the material satisfies the constitutive Equations (18) and (41). This is because the regime of flow at the interface is completely described by the constitutive equations. This feature of solutions is typical for several rigid plastic models [33].

It is, in general, assumed that the shear stress is approximately constant inside the sample if the angle $\beta$ (Figure 1) is small enough [4,11]. A consequent conclusion is that the shear strain rate is also approximately constant. This conclusion is not valid for the material models satisfying (18) at $\alpha < 0$.

Indeed, in this case the shear strain rate may approach infinity at one of the contact surfaces, as it is seen from (39).

The models that do not satisfy (17) have not been considered in the present paper. However, such models are widely used in applications, for example, the Herschel-Bulkley model [34]. It has been shown in [35] that it is always possible to find a solution satisfying the regime of sticking for such models. Therefore, the qualitative behavior of solution for the models that do not satisfy (17) is similar to that for the models satisfying (18) if the inequality in (40) is valid. It is believed that this qualitative difference provides a means for testing different models experimentally. However, it is necessary to design a special experimental program. The function introduced in (18) does not affect the behavior of solutions outside a very narrow layer near the friction surface and this function cannot be determined from standard tests. This is because the strain rate is bounded in any test. Let $\xi_{max}$ be the maximum possible shear strain rate in experiment. Then, Equation (15) can be rewritten as

$$
\frac{\sigma_{\theta\varphi}}{k_0} = \begin{cases} \Phi\left(\frac{\xi_{\theta\varphi}}{\xi_0}\right) \text{ if } \xi_{\theta\varphi} \geq \xi_{max} \\ \Phi_0\left(\frac{\xi_{\theta\varphi}}{\xi_0}\right) \text{ if } \xi_{\theta\varphi} \leq \xi_{max} \end{cases}
\tag{48}
$$

where $\Phi_0\left(\xi_{\theta\varphi}/\xi_0\right)$ is an arbitrary function of its argument. This function can be chosen to approximate the actual experimental data in the range $0 < \xi_{\theta\varphi}/\xi_0 < \xi_{max}/\xi_0$ with any desirable/optimal accuracy. Then, the function $\Phi\left(\xi_{\theta\varphi}/\xi_0\right)$ can be chosen to satisfy (18) with any value of $\alpha$. It is also possible to choose this function such that (17) is not satisfied. A requirement is that $\Phi(\xi_{max}/\xi_0) = \Phi_0(\xi_{max}/\xi_0)$ and $\Phi'(\xi_{max}/\xi_0) = \Phi'_0(\xi_{max}/\xi_0)$. It is also possible to require that higher derivatives are continuous. The choice of the function $\Phi\left(\xi_{\theta\varphi}/\xi_0\right)$ does not affect the accuracy of approximation of the actual experimental data but affects the qualitative features of the theoretical solution. Some of these features are feasible for experimental verification using indirect methods. In metal plasticity, it has been already recognized that standard tests cannot be used to identify the constitutive equations in a narrow layer near frictional interfaces [36].

## 6. Conclusions

Presented herein is a general semi-analytic solution for material flow in the cone-and-plate rheometer assuming quite a general constitutive equation that connects the shear stress and shear strain rate. This equation involves quite an arbitrary function $\Phi\left(\xi_{eq}/\xi_0\right)$, as shown in (7). Then, attention is focused on a class of the functions satisfying (18). The asymptotic analysis carried out has demonstrated that the qualitative behavior of the solution essentially depends on the parameter $\alpha$. In particular, if this parameter satisfies the inequality (41) that the regime of sliding occurs at some angular velocity independently of the quality of the surface and other conditions. This property of the solution can be used for experimental verification that the material satisfies the constitutive Equation (18) with $\alpha$ satisfying (41). Moreover, it is seen from (39) that if $\alpha < 0$ then the shear strain rate approaches infinity in the vicinity of the surface $\theta = \theta_0$ (Figure 2). This mathematical property of the solution is not feasible for direct experimental verification. However, it is well known that a narrow layer with vanishing viscosity is usually generated near rigid walls [28]. This is an indirect confirmation that the shear strain rate is very high within this layer and that the gradient of the shear strain rate in the direction normal to the wall is also high.

The solution is based on typical assumptions adopted in theoretical analyses of rheometers. It is known that these assumptions are not always adequate [3,37]. A numerical technique is required to find a solution without the assumptions. However, this possible numerical solution may be quantitatively different from the semi-analytic solution found but the main qualitative features of the semi-analytic solution near the rigid wall are independent of other boundary conditions and assumptions. Therefore, any numerical solution should have all qualitative features inherent to the semi-analytic solution. In particular, the regime of sliding should occur at some value of the angular

velocity of the cone. Moreover, the solution should be singular at the rigid wall if the regime of sliding occurs. The latter greatly adds to the difficulties of numerical solutions. In particular, traditional finite element methods are not capable of dealing with this kind of problem.

The solution found is for the cone-and-plate rheometer. It is evident from the method of finding the solution that it can be easily extended to the cone-and-cone rheometer, which is also often used for determining the properties of viscoplastic materials [4,38].

**Author Contributions:** Conceptualization, S.A.; writing, L.L.; formal analysis, E.L. and V.M.D. All authors have read and agreed to the published version of the manuscript.

**Funding:** This research was made possible by grants RFBR-17-51-540003 and VAST.HTQT.NGA.07/17-18.

**Conflicts of Interest:** The authors declare no conflict of interest.

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
