# Peer review of "The Behavior of Melts with Vanishing Viscosity in the Cone-and-Plate Rheometer"

_applsci, doi:10.3390/app10010172_

Round 1

Reviewer 1 Report

Review of The Behavior of Melts with Vanishing Viscosity in the Cone-and-Plate Rheometer

The work has a good value and can be published after a minor revision. I think that the following comments will help authors to reconsider how they "package" their results. It is a nice point of view that a nonlinear viscosity relation is solving the problem with the boundary layer; however, this outcome is not stated in the paper clearly. Only someone being able to follow the mathematics would be seeing this outcome, yet the paper should be written for everyone, especially people doing inverse analysis or even computer simulations. I think that a minor revision will increase the likelihood to get a better attention.

1. Please rewrite the Abstract in order to make sentences shorter, to the point, and also diminish repetitive use of "that" in the same sentence.

2. The authors discuss the system by using/motivating an analytical solution and then write this lax statement: "In the case of small cone angles, the flow in the cone-and-plate rheometer is considered to be ideal in the sense that it is practically homogeneous." I would invite them simply use their solution in order to see how good this estimate really is, if they want to. The error for cone angle 4 degrees is less than 0.5%, so this "consideration" is very accurate, not simply an assumption without any justification.

3. It is perfectly adequate to neglect normal stress effects (Weissenberg effect) and set sigma_theta=sigma_r, but this assumption has to be stated and the paper has to restricted to such materials.

4. I believe that authors will benefit from the book, especially in introducing concepts like components and physical coordinates in spherical coordinates: Coleman, B. D., Markovitz, H., and Noll, W., Viscometric flows of non-Newtonian fluids: theory and experiment, vol. 5 (Springer-Verlag, 1966) I also advise to cite more recent works to get attracted from active rheologists, even in this journal Applied Sciences MDPI searching for rotational rheometer shows works in this direction.

5. An important fact, Eq. (16) is correct only if the paper is talking about physical coordinates. There is no clarity about that and it looks like that covariant components of tensors are used. Please be precise.

6. Indeed it is not clear, if the issue is about the homogeneous stress or strain rate. I believe that the Introduction is not clear about the real study, which becomes a bit more clear at the Eq. (16) and then the statement eta goes to zero whence velocity gradient approaches infinity. This case occurs for a homogeneous stress even within the boundary layer. So the paper is actually interested how the justification of solution dependence on theta is verified. Please kindly add a drawing showing how velocity gradient goes to infinity, by using a asymptotic boundary layer. Otherwise, it is simply mathematics and not clear why it is even possible to think that stress remains constant where velocity gradient approaches infinity.

7. After (19) the terminology "rate-independent material" is confusing, use a linear material or a constant viscosity or "rate-independent viscosity" might be the wording.

8. The work has to state its own sentence in the Introduction rather than on page 4 "Therefore, in contrast to conventional approaches, the transition between the regimes of sticking and sliding is controlled by the constitutive equations rather than by the friction law."

9. Please kindly bring the Conclusion with Herschel--Bulkley equation in connection and provide a basis for experimentalists why they should use such a nonlinear viscosity by using the argumentation in your work. It is a nice study, but not really sharp enough to get its potential value. Herschel, W.H.; Bulkley, R. (1926), "Konsistenzmessungen von Gummi-Benzollösungen", Kolloid Zeitschrift, 39: 291–300

Author Response

Please see the file attached. The corrections made in the manuscript are shown in red.

Reviewer 2 Report

Contrasting to the name of the journal ”Applied Sciences“ the whole manuscript is purely theoretical with no immediate impact to constitutive modelling of the experimental rheological data. The following points should be more addressed:

The approach is consecutively constructed rather artificially with the assumptions enabling further steps (monotonicity of the function Φ (p.3, l.173 + 5-113), a form of function Φ (rel. (18)), etc.). The a priori assumption of neglecting inertia effects (2-51) should be justified in combination with ω®∞ (5-125). The authors frequently repeat: “the possibility to control the occurrence of slip using appropriate constitutive equations in the vicinity of the wall...” (1-44,45); “the transition between the regimes of sticking and sliding is controlled by the constitutive equations...” (4-106,107); “the regime of flow at the interface is completely controlled by the constitutive equations...” (8-191,192).

However, the primary input is represented by the experimental data and any constitutive models (empirical, phenomenological, differential, integral) should serve to their evaluation and not on the contrary. No constitutive model can control behaviour of the studied material.

8-206 “independently of the quality of the surface”

The solution is derived in the spherical coordinates in which the surfaces of a plate and a cone are strictly described and hence, the presented solution strictly supposes an absolute smoothness of both surfaces. Any deviation of any surface from an absolute smoothness cannot be described in the spherical coordinates and in this case no solution is at disposal.

The authors provide no application to the real rheological data to verify their conclusions. The conclusions are based on the solution “generated” by the consecutive assumptions. Nevertheless, there exist more possibilities and from the selected one is rather difficult to come to the description of flow behaviour with no concrete practical verification. If the function Φ would be expressed through other functional description, different conclusions will be derived and the question is: which approach is more useful in connection with data processing and modelling?

The authors should substantiate their approach in more detail.

Author Response

(The authors gave the same response as above.)

Round 2

Reviewer 2 Report

The authors’ approach to the manuscript is clear. However, it seems me that my recommendations concerning an improvement of their presentation were not understood. My intention is not to oppose their results as their procedure seems to be good, but it should be useful to elucidate their approach in two following aspects:

1) Flow behaviour (experimental findings) should be ‘superordinate’ to its description that ‘only’ tries to describe material behaviour

2) It should be useful to ‘interlace’ the material and mathematical terms (assumptions) otherwise an impact to practical application is substantially attenuated.

Ignoring these two aspects limits applicability of the presented results into practice, especially as no example to real data is provided.

In this sense the points raised in the preceding review are still valid and are also documented by the following examples:

l. 49 - To mention viscous and viscoelastic materials - to interlace mathematical assumptions with a characterization of non-Newtonian materials

l. 99 - “It is evident that this class is large enough for any applications.” - E.g. in polymer processing decisively not.

l. 101 - “Since Φ is a monotonically increasing function of its argument...” - A suitable place to justify an assumption of monotonicity of Φ in the sense that it covers ...

l. 115 - “the transition between the regimes of sticking and sliding is controlled by the constitutive equations rather than by the friction law.” - Would not be better to replace the word ‘controlled’ by the word ‘described’? The physics are a primary source, modelling the physical phenomena is secondary and subjects to physics.

l. 200 - “It is because the regime of flow at the interface is completely controlled by the constitutive equations.” - The CEs serve for modelling of flow behaviour, in no case they can control flow behaviour.

l. 219 - “This function can be chosen to approximate the actual experimental data in the range ... with any accuracy.” - It is not possible. Any experimental data are inaccurate. Measurement of shear viscosity ranges to the most accurate measurements (deviation up to 5 %?). Nevertheless, there is always a ‘non-uniform’ deviation from an optimal course. Such data cannot be modelled with any accuracy.

Author Response

Please see the file attached

Round 3

Reviewer 2 Report

In this version the authors attenuated dominance of mathematics over physics, which was my principal comment concerning their version 2.

For improvement:

l. 83 - increases